# Species-specific maturation profiles of human, chimpanzee and bonobo neural cells

Maria C Marchetto[1†]*, Branka Hrvoj-Mihic[2†], Bilal E Kerman[3†], Diana X Yu[4], Krishna C Vadodaria[1], Sara B Linker[1], Iñigo Narvaiza[1], Renata Santos[1,5], Ahmet M Denli[1], Ana PD Mendes[1], Ruth Oefner[1], Jonathan Cook[1], Lauren McHenry[1], Jaeson M Grasmick[1], Kelly Heard[1], Callie Fredlender[1], Lynne Randolph-Moore[1], Rijul Kshirsagar[1], Rea Xenitopoulos[1], Grace Chou[1], Nasun Hah[1], Alysson R Muotri[6,7], Krishnan Padmanabhan[8], Katerina Semendeferi[2,9]*, Fred H Gage[1]*

[1]Laboratory of Genetics, The Salk Institute for Biological Studies, La Jolla, United States; [2]Department of Anthropology, University of California, San Diego, La Jolla, United States; [3]Regenerative and Restorative Medicine Research Center (REMER), Istanbul Medipol University, Istanbul, Turkey; [4]Department of Oncological Sciences, Huntsman Cancer Institute, Salt Lake City, United States; [5]Laboratory of Dynamic of Neuronal Structure in Health and Disease, Institute of Psychiatry and Neuroscience of Paris (UMR S894 INSERM, University Paris Descartes), Paris, France; [6]Department of Pediatrics, University of California, San Diego School of Medicine, La Jolla, United States; [7]Department of Cellular & Molecular Medicine, Rady Children's Hospital San Diego, La Jolla, United States; [8]Department of Neuroscience, University of Rochester School of Medicine and Dentistry, Rochester, United States; [9]Neuroscience Graduate Program, University of California San Diego, La Jolla, United States

*For correspondence:
marchetto@salk.edu (MCM);
ksemende@ucsd.edu (KS);
gage@salk.edu (FHG)

†These authors contributed
equally to this work

Competing interest: See
page 19

Reviewing editor: Paola Arlotta,
Harvard University, United States

**Abstract** Comparative analyses of neuronal phenotypes in closely related species can shed light on neuronal changes occurring during evolution. The study of post-mortem brains of nonhuman primates (NHPs) has been limited and often does not recapitulate important species-specific developmental hallmarks. We utilize induced pluripotent stem cell (iPSC) technology to investigate the development of cortical pyramidal neurons following migration and maturation of cells grafted in the developing mouse cortex. Our results show differential migration patterns in human neural progenitor cells compared to those of chimpanzees and bonobos both in vitro and in vivo, suggesting heterochronic changes in human neurons. The strategy proposed here lays the groundwork for further comparative analyses between humans and NHPs and opens new avenues for understanding the differences in the neural underpinnings of cognition and neurological disease susceptibility between species.
DOI: https://doi.org/10.7554/eLife.37527.001

## Introduction

Since the split from the last common ancestor of hominins (*Homo*) and African apes (*Gorilla* and *Pan spp*), human brain evolution has been characterized by several waves of increases in cranial capacity (*Carlson et al., 2011*; *Falk et al., 2000*) and selective expansion of regions implicated in complex cognition (*Semendeferi et al., 2010*; *Semendeferi and Damasio, 2000*). The increase in the cranial

capacity in fossil hominins has been tied to behavioral changes, including the appearance of the first stone tools and their subsequent elaboration, increases in population size, and the spread of hominins into ecologically challenging habitats (*Ambrose, 2001*; *Stout, 2011*). Equally important were the subtle changes in brain organization at the microscopic level. While these changes cannot be inferred directly from the fossil specimens, comparative analyses of cortical organization between extant primate species suggest that the human brain indeed differs from the brain of other hominid species in several important microstructural aspects (*Semendeferi et al., 2001*; *Barger et al., 2007*; *Semendeferi et al., 2011*). One of these is the dendritic morphology of cortical pyramidal neurons, which differs between humans and the common chimpanzee (*Pan troglodytes*) (*Bianchi et al., 2013a*). Since cortical pyramidal neurons represent the most common type of neuron in the cortex and form basic units of cortical microcircuitry (*DeFelipe and Fariñas, 1992*), comparative analyses directed specifically at pyramidal neurons can yield insights into the organization of the microcircuitry that is typical of each species.

Differences observed in the adult phenotype between humans and chimpanzees likely reflect differences in timing and/or rate of cortical development. Developmental differences between species represent an important component in evolutionary studies, as small changes in the timing of development translate into morphological differences in adulthood, often with important functional implications. In the case of human brain evolution, sequential hypermorphosis, a type of heterochrony characterized by prolongation of all stages of brain development compared to the ancestral state (*McNamara, 2002*; *Vrba, 1998*; *McKinney, 2002*), has been proposed as an evolutionary mechanism underlying cerebral expansion in humans. Humans and apes share a pattern of prolonged postnatal growth in brain size that sets them apart from Old World monkeys (*Leigh, 2004*). Similarly, the maturation of pyramidal neurons appears to be protracted in hominids compared to macaques (*Cupp and Uemura, 1980*; *Petanjek et al., 2011*; *Sedmak et al., 2018*) and in humans compared to chimpanzees (*Bianchi et al., 2013a*; *Teffer et al., 2013*), possibly accounting for longer and more branched dendrites with higher numbers of dendritic spines in humans (*Bianchi et al., 2013b*; *Petanjek et al., 2008*). Because there is little evidence that the simple addition or subtraction of genes is sufficient to explain the observed differences (*Hill and Walsh, 2005*), changes in the regulation (levels and patterns of expression) of genes shared between humans and chimpanzees have been proposed to play an important role (*King and Wilson, 1975*; *Enard et al., 2002*). In particular, the expression of genes relevant to developmental events such as migration and dendritic maturation in pyramidal neurons can provide important insights into the underlying mechanisms shaping the differences in the organization of neuronal networks observed in humans and other primates.

Although studies using post-mortem brains from human and NHPs have provided important insights into developmental differences across species, the availability of specimens often limits the extent of the hypotheses that can be addressed. Recent advances in somatic reprogramming technology make comparative studies possible even in the absence of post-mortem specimens (*Marchetto et al., 2013*). The work presented here specifically addresses neural progenitor cell (NPC) migration and the development and functional maturation of cortical neurons in humans and chimpanzees. While changes in neural migration and cortical layering may pathologically affect early dendritic organization and microcircuitry formation in humans (*Pramparo et al., 2015*; *Reiner et al., 2016*; *Muraki and Tanigaki, 2015*; *Brennand et al., 2015*), previous research has not examined migration and early development of pyramidal neurons from an evolutionary perspective. We therefore hypothesized that the neurodevelopmental differences between humans and chimpanzees would become visible during distinct processes of NPC migration and during the initial establishment of the organization of dendritic trees and functional neuronal maturation in the neocortex. To test this hypothesis, we utilized induced pluripotent stem cell (iPSC) technology to model NPC migration in *Homo* and *Pan spp.* (chimpanzee and bonobo) and in the early development of cortical neurons.

We found differential migration patterns in human NPCs compared to those of chimpanzee and bonobo based on RNA expression profile analysis and live-cell imaging. Next, we observed morphological and functional developmental differences between human and chimpanzee neurons, suggesting differences in the timing of neuronal maturation between the two species. We report here in vitro and in vivo comparative analyses of the neural development of two closely related primate species. The strategy applied in this work can be utilized for further studies addressing human brain

evolution and the mechanisms underlying the cellular and molecular aspects that are unique to the human brain.

## Results

### Analysis of the expression profiles of human, chimpanzee and bonobo NPCs shows differentially regulated genes related to cell migration

Fibroblasts from chimpanzees (*Pan troglodytes*), bonobos (*Pan paniscus*) and humans (*Homo sapiens*) were reprogrammed to their pluripotent state as previously described (*Brennand et al., 2015*) and then differentiated into NPCs. Briefly, we initiated neural differentiation by plating neuralized embryoid bodies (EBs) for a week until neural rosettes became apparent in the dish. Dissociated rosettes formed a homogeneous population of NPCs that continued to proliferate in the presence of the mitogen FGF2 and were positive for the pan-neural progenitor marker Nestin (*Figure 1A and B*) and cortical neural progenitor markers Pax6 and Foxg1 (*Figure 1—figure supplement 1A–C*). To determine whether there was a difference in cortical layer identity between species at the population level, we performed single cell RNA sequencing on NPCs derived from a human, bonobo, or chimpanzee sample. The results revealed that each species had similar proportions of cortical markers, indicating that within the cortical-fated cells there was no enrichment for any given layer between the species (*Figure 1—figure supplement 2A–C*, see Materials and methods for details).

We also investigated whether human and *Pan* NPCs were similarly specified to cortical and hindbrain fates. For this analysis, we determined the proportion of NPCs expressing well-known cortical and hindbrain markers (*Figure 1—figure supplement 2C*) SOX2 and NESTIN expression were used to identify NPCs; PAX6, NEUROD6 and TBR1 to identify cortical NPCs; and EN1, EN2, ISL1, and LHX3 to identify hindbrain NPCs. We identified similar proportions of all NPCs, cortical NPCs, and hindbrain NPCs in all species. This analysis further indicated that there was no detectable species-dependent bias in regional patterning.

To gain insights into differences in gene expression between human, chimpanzee and bonobo NPCs, we performed high-throughput RNA sequencing (RNA-seq) analyses on eight human and five nonhuman primate (chimpanzee and bonobo) NPC lines. We assessed the RNA-sequencing data from NPCs for developing brain regional signatures. Using *BrainImageR* tool (*Linker et al., 2019*), we calculated the enrichment of each differentially expressed gene (described on *Supplementary file 1*) within tissues of the developing brain. Genes upregulated in *Homo* NPCs exhibited similar regional enrichment to genes upregulated in *Pan* (chimpanzee and bonobo) NPCs, indicating that the differences observed in RNA sequencing were not influencing broad fate specification between NPCs from *Homo* versus *Pan* (*Figure 1—figure supplement 1D*). The expression profiles of NPCs from chimpanzee and bonobo clustered closer to each other than to human NPC lines derived from iPSC or human embryonic stem cell (ES) (*Figure 1C–E*). The Venn diagram in *Figure 1D* represents protein-coding expressed genes with non-significant differences between species, differentially regulated genes with estimated false discovery rates (FDR) of less than 5% and a fold change greater than 2-fold (434 genes for human and 762 genes for chimpanzees and bonobos). Importantly, a principal component analysis (PCA) plot confirmed that human samples clustered separately from nonhuman primates (*Pan*) unbiasedly. The first principal component (33% of the variance) separated human from chimpanzee and bonobo samples (*Figure 1E*). We next focused on genes differentially expressed between *Homo* and *Pan* NPCs and found that, among the 1196 differentially expressed genes (*Supplementary file 1*), 52 genes were involved in pathways related to cellular migration (Fisher's exact test, p<1.07e-24) (*Figure 1F*, *Figure 1—figure supplement 1E* and *Supplementary file 2* and *3*).

### Differential migration properties in human compared to chimpanzee and bonobo NPCs in vitro

We performed a scratch assay, that is wound healing assay, (see Materials and methods) to further evaluate the migration properties of NPCs (*Liang et al., 2007*). In *Figure 2A*, representative images taken at the time of the scratch (0) and after 12 and 24 hr for the different species are shown. While both chimpanzee and bonobo NPCs were confluent by 24 hr, a gap remained at 24 hr in the human NPCs (*Figure 2A*, bottom panels). To better explore the migration differences between species, we

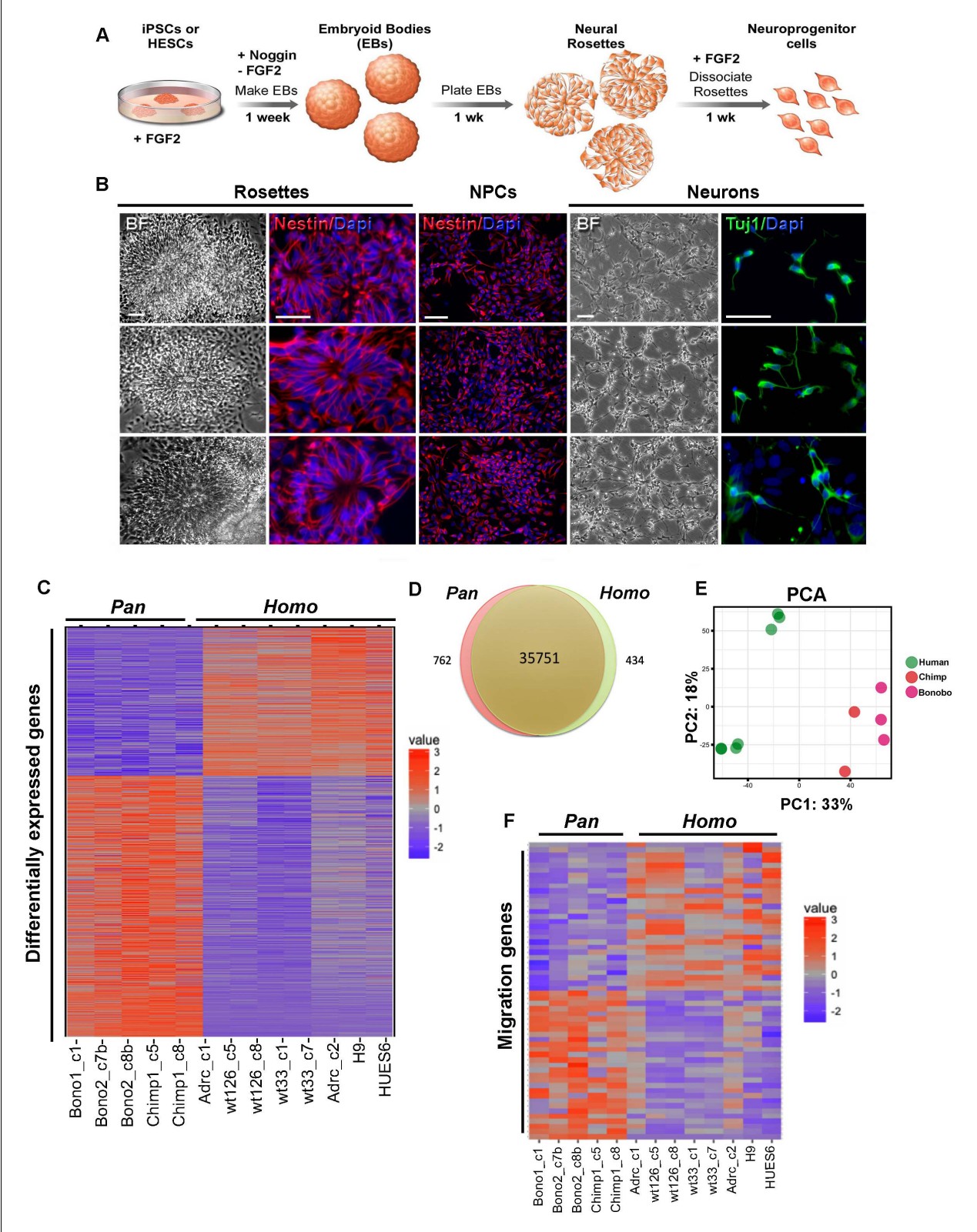

**Figure 1.** Characterization of neural progenitor cells (NPCs) derived from human (*Homo*), chimpanzee and bonobo (*Pan*) iPSCs. (**A**) Scheme showing the main steps and timeline involved in the generation of NPCs from iPSCs. (**B**) Morphological aspects of neural rosettes derived from embryoid bodies (EBs) (BF: Bright Field). Neural rosettes are positive for the early neural marker Nestin (red). In the presence of FGF2, dissociated rosettes form a homogeneous monolayer population of proliferating Nestin-positive NPCs. Upon FGF2 withdrawal, NPCs differentiate readily in Tuj-1-positive neurons

*Figure 1 continued on next page*

*Figure 1 continued*

(green). Scale bars: 50 µm. (C) Heat map representation of mapped reads corresponding to protein coding genes. Chimpanzee and bonobo NPCs clustered closer to each other than to human NPC lines and to human embryonic stem cell-derived NPCs (HUES6 and H9 ES lines). (D) Venn diagram showing pairwise comparison of protein-coding genes. Red or green: significantly differentially expressed genes (FDR < 0.05 and fold change >2). Red and green: total of expressed genes with no significant differences in mRNA levels between compared species. (E) Principal component analysis (PCA) of human, chimpanzee and bonobo NPCs showing human samples (green circles) clustering separately from NHPs (red and pink circles) on the first principal component (33% of the variance). (F) Heat map showing comparison between *Homo* and *Pan* NPC expression profiles of genes related to cell migration.

DOI: https://doi.org/10.7554/eLife.37527.002

The following figure supplements are available for figure 1:

**Figure supplement 1.** Similar expression of cortical progenitor markers and similar spatial enrichment of differentially regulated genes in *Homo* and *Pan*.

DOI: https://doi.org/10.7554/eLife.37527.003

**Figure supplement 2.** Single cell analysis on human and *Pan* NPCs.

DOI: https://doi.org/10.7554/eLife.37527.004

quantified two features of NPC migration: 1) the path taken by NPCs in response to the scratch and 2) the dynamics of the cells as they moved along that path. Individual NPCs were randomly selected from the population in each experiment and tracked manually (*Figure 2—figure supplement 1A*) for the three species. A total of 340 human, 165 chimpanzees, and 186 bonobo NPCs were analyzed from 57 human, 36 chimpanzee, and 34 bonobo movies (see Materials and methods). Representative cell tracks for each species are shown (*Figure 2B–D*). The starting point for each example cell in the track was marked with a colored circle (*Figure 2B*, human = black, chimpanzee = red and bonobo = blue), the path taken was marked with a colored line and the final location was marked with a gray circle. To test if the cells migrated toward the scratch, the distance difference (DD) for each NPC of each species was calculated and compared to the DD for simulated random migration. For all three species, migration movement was directed toward the scratch when compared to random migration (*Figure 2—figure supplement 1B–D*). Therefore, we determined that the scratch assay can be used to compare directional migration of NPCs from humans, chimpanzees and bonobos.

Next, we characterized features of the migration process itself and focused on the speed of NPC migration (*Figure 2*). For single NPCs, we first characterized the speed at each frame with respect to both space (*Figure 2—figure supplement 1E–G*) and over time (*Figure 2—figure supplement 1H–J*). Although cells from all species showed periods of slow migration (between 0 µm /min and 1 µm /min), only chimpanzee and bonobo NPCs had periods of rapid migration (1.25 µm /min to 2.5 µm /min). As a result, the distribution of speed for chimpanzee and bonobo NPCs had a heavy tail corresponding to these periods of rapid migration. To compare the speed profiles across the three species, we first calculated the cumulative distance covered by each cell and the mean cumulative distance covered for each species and calculated the migration speed per cell (*Figure 2E,F*). The distribution of migration speed of cells from the three species differed, with human NPCs moving significantly more slowly than either chimpanzee or bonobo NPCs (mean migration speed: human = 0.46 ± 0.19 µm /min, chimpanzee = 0.70 ± 0.31 µm /min, bonobo = 0.72 ± 0.35 µm /min, *Figure 2F*). By contrast, we did not find significant differences in the migration speeds between chimpanzee and bonobo NPCs.

We next confirmed the cell migration differences between species by performing an independent assay, the neurosphere outgrowth assay. We measured the distance that NPCs migrated radially from a plated neurosphere. We compared outgrowth from human-, chimpanzee- and bonobo-derived spheres generated by culturing NPCs in non-adherent conditions for 48 hr and then plating them on coated plates, as previously described (*Brennand et al., 2015*; *Delaloy et al., 2010*). We observed significantly reduced outgrowth from human neurospheres (331.7 ± 14.24 µm, total of 123 neurospheres analyzed) relative to chimpanzee and bonobo (392.7 ± 18.86 µm for chimpanzees and 404.3 ± 20.98 µm for bonobos; total of 135 neurospheres analyzed) (*Figure 2G and H*). To evaluate if differential migratory properties were influenced by the presence of cells from other brain regions due to loss of cell purity during migration, we post-stained human and NHP cells after neurosphere migration experiment with cortical progenitor marker (PAX6) and neuroblast migration marker

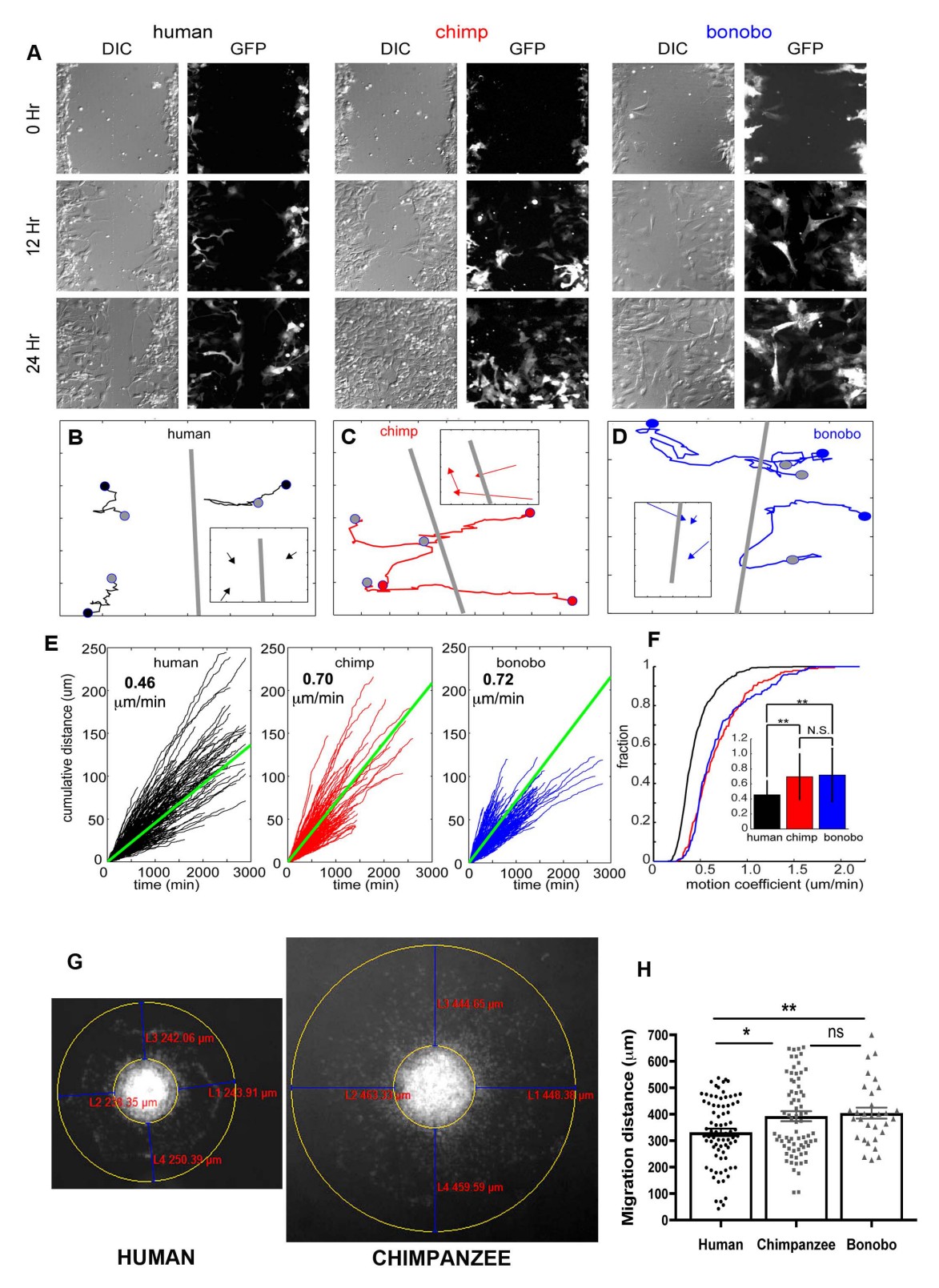

**Figure 2.** NPC differential migration patterns in humans, chimpanzees and bonobos. (A) DIC (differential interference contrast) (left) and fluorescent (GFP) (right) images of time-lapse NPC migration from humans, chimpanzees, and bonobos at 0 hr (top row), 12 hr (middle row), and 24 hr (bottom row) following a scratch made with a glass pipette in the center of the field of view (at 0 hr). (B–D) Individual tracks with NPC origin marked with a black circle for human NPCs (B), red circle for chimpanzee NPCs (C), and blue circle for bonobo NPCs (D). Migration paths for each NPC are marked with a

*Figure 2 continued on next page*

*Figure 2 continued*

line and the end of the migration is marked with a gray circle. The scratch location is marked with a gray line. Inserts contain vectors for each cell showing net migration (from beginning of the track to the end – colored arrows for each NPC) relative to the scratch (gray line). (E) Cumulative distance from each NPC over time from all human (black lines), chimpanzee (red lines) and bonobo (blue lines). Also shown is the mean cumulative distance for each species (green line). (F) Cumulative histogram of motion coefficients of NPCs from three species and mean motion coefficient for NPCs from three species (inset) (**$p<0.01$, Mann-Whitney U test), showing that chimpanzee and bonobo NPCs migrate longer distances at higher speeds compared to human NPCs. (G–H) Migration analysis using neurosphere outgrowth assay. (G) Representative images of iPSC forebrain neurospheres from human and chimpanzee showing differential outgrowth migration of NPCs. The average distance between the radius of the inner neurosphere and outer circumference of cells was calculated using (DAPI)-stained nuclei (blue). Four measurements were taken and averaged for each neurosphere (examples of distances shown in red). (H) Graph showing quantification of distances migrated by human, chimpanzee and bonobo NPCs. The data confirmed that NPCs from chimpanzees and bonobos migrate longer distances than human NPCs. Comparative statistics as follows: Human *versus* Chimpanzee, $p=0.0058$; Human *versus* Bonobo, $p=0.0108$; Chimpanzee *versus* Bonobo, $p=0.6831$. *$p<0.05$, **$p<0.01$, Welch's t test).

DOI: https://doi.org/10.7554/eLife.37527.005

The following figure supplement is available for figure 2:

**Figure supplement 1.** Characterization of migration patterns on NPCs.

DOI: https://doi.org/10.7554/eLife.37527.006

(doublecortin, DCX). We did not observe significant differences between species, even though the migration distances covered by human NPCs were significantly shorter than the distances covered by NHP cells (*Figure 2—figure supplement 1K–M*). In conclusion, individual NHP NPCs reached higher instantaneous speeds, had higher migration speeds and migrated further than human NPCs. Taken together, differences in migration speed and neurosphere outgrowth demonstrated that human NPCs migrated more slowly in vitro than chimpanzee and bonobo NPCs.

## Differential migration properties in human compared to chimpanzee NPCs in vivo

We then asked if the difference seen in the migration speeds was influenced by the in vitro culture conditions or represented a *bona fide* difference in underlying developmental programs that have changed during evolution. To address this question, we grafted NPCs from two human and two chimpanzee NPC lines into the developing postnatal mouse cortex (*Figure 3A*). Chimpanzee and human NPCs expressed different fluorescent markers (GFP and TdTomato) and were mixed together at a 1:1 ratio prior to transplantation. Mice were sacrificed at two weeks post-transplantation (PST) and two different migration parameters were analyzed: (1) the area covered by migrating NPCs and (2) the distance migrated by NPCs from the injection site (*Figure 3B,C and D* and *Figure 3—figure supplement 1A,B*). Once again, chimpanzee NPCs showed an increase in area coverage and distance traveled from injection site. In summary, our data show that chimpanzee and bonobo NPCs migrated faster and covered longer distances than human NPCs, independent of culture conditions. We conclude that faster migration represents a cell autonomous feature of NHP NPCs that could influence the differences in brain size and brain organization between species.

## Differential neuronal maturation of human and chimpanzee cells after transplantation

Neuronal progenitor migration and neuronal maturation are distinct processes that could be interconnected; therefore, we chose to compare human and chimpanzee neuronal differentiation in vivo by looking at morphological characteristics that are indicative of neuronal developmental steps. To provide an enriched niche and environmental cues for proper neuronal development, we grafted NPCs from two human and two chimpanzee lines into the developing postnatal mouse brain (see Materials and methods for details). Mice were sacrificed at different time points post transplantation (PST), and the morphological characteristics of transplanted neurons were quantified (*Figure 4*, *Figure 4—figure supplements*). The selected duration of in vivo neuronal growth was 19 weeks, which, based on human post-mortem studies, spans most of the period of prenatal development of pyramidal neurons in the human cortex, that is from the emergence of the first cells with pyramidal morphology at 17 weeks gestation until birth (*Mrzljak et al., 1988*). Neurons with pyramidal morphology were identified and traced at each time point of the study. They were distinguishable by their large soma, neurites emerging from the cell body, and visible dendritic spines

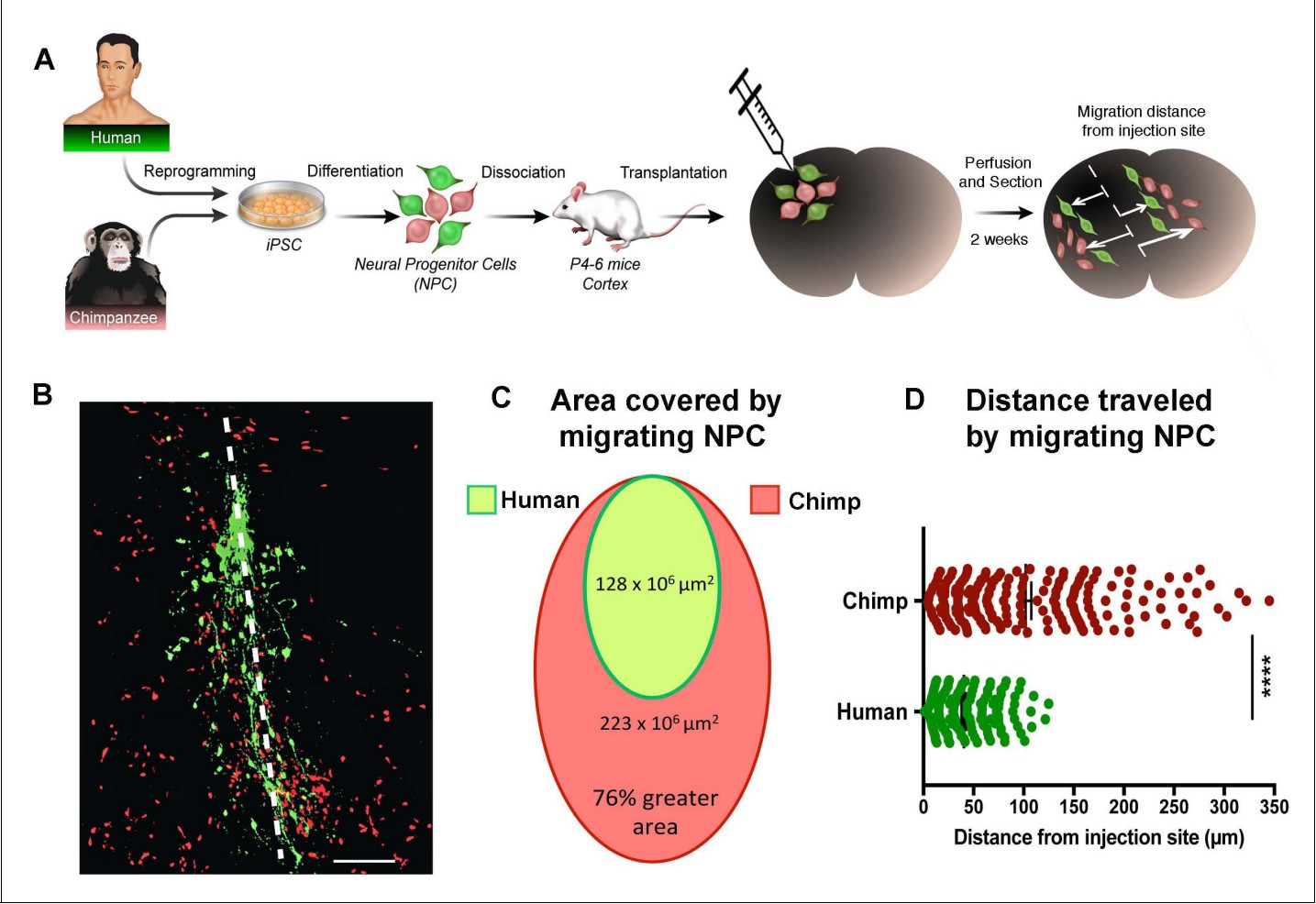

**Figure 3.** In vivo migration comparative analysis. (A) Scheme of the transplantation experimental procedure to detect in vivo cell migration. Fibroblasts from human and chimpanzee were reprogrammed to iPSCs and differentiated to NPCs and then transduced with lentiviral vectors expressing either green fluorescent protein (GFP) or TdTomato. NPC lines from human and chimpanzee were mixed and grafted into the developing postnatal mouse cortex. Mice were sacrificed at two weeks post-transplantation. (B) Representative image of grafted human (green) and chimpanzee (red) NPCs migrating away from the injection site (indicated by the white dashed line). (C) Representation of the area covered by human and chimpanzee migrating NPCs (D) Distance migrated by single NPCs from the injection site. (****p<0.0001, Mann-Whitney U test).

DOI: https://doi.org/10.7554/eLife.37527.007

The following figure supplement is available for figure 3:

**Figure supplement 1.** In vivo migration of human and chimpanzee NPCs two weeks after transplantation in mouse brain.

DOI: https://doi.org/10.7554/eLife.37527.008

---

(*Figure 4B*). We identified pyramidal neurons distributed across all cortical layers, in callosal fibers and clustered around ventricles; however, given that the focus of this study is the development of cortical neurons, only the neurons with cell bodies located in the cortex and corpus callosum were included in the analysis (*Figure 4B,C*).

## Transplanted iPSC-derived neurons show changes in dendritic features over time

We analyzed changes in dendritic parameters over time to investigate dendritic development. Specifically, we measured total dendritic length (TDL), mean segment length (MSL), dendritic spine number (DSN), dendritic spine density (DSD), dendritic segment count (DSC), number of dendrites (TREE), and cell body area (SOMA) (*Figure 4D–K*).

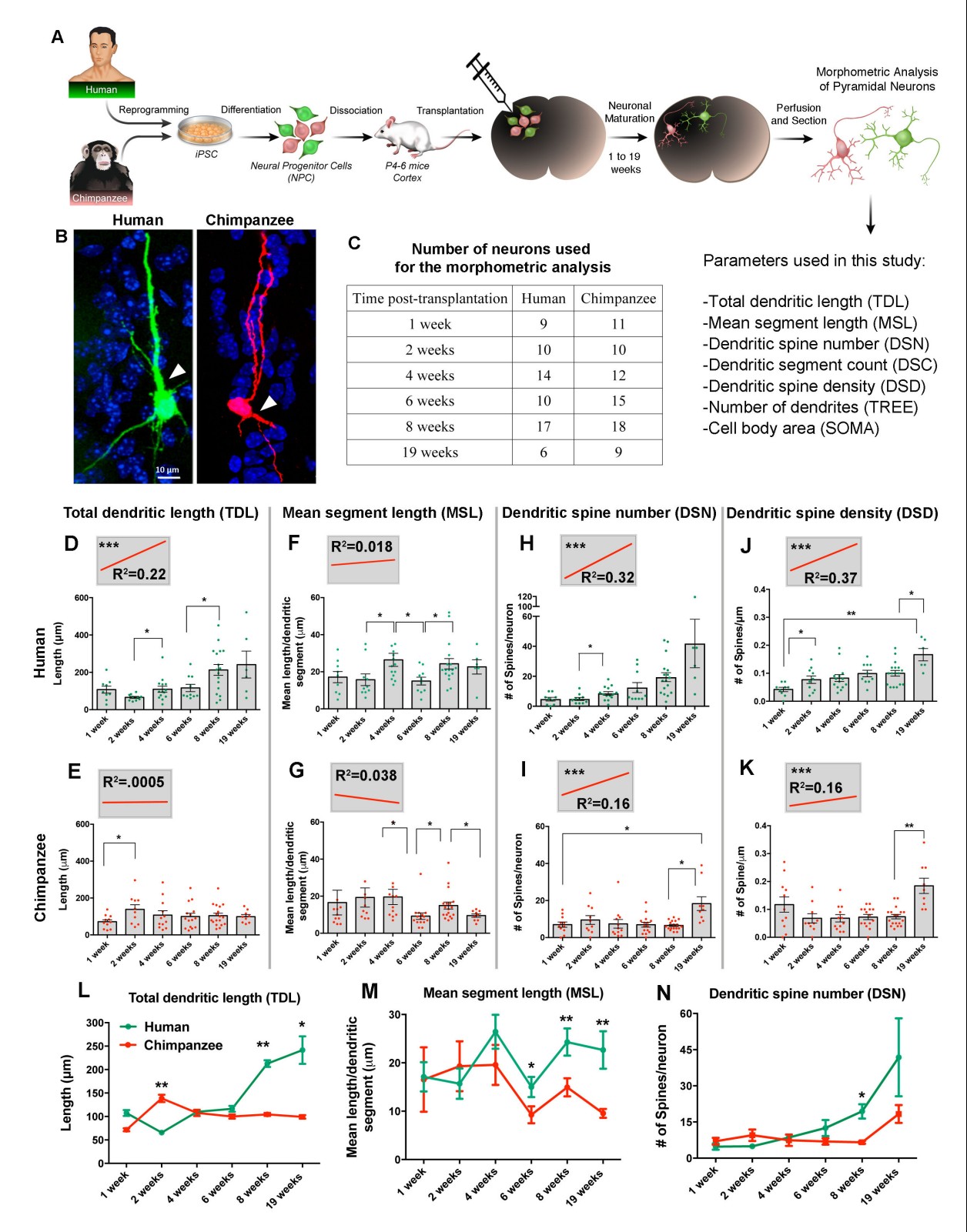

**Figure 4.** Comparative morphometric analyses on iPSC-derived transplanted neurons over time. (**A**) Scheme of the transplantation experimental procedure. NPCs were obtained from both species and transduced with lentiviral vectors expressing either green fluorescent protein (GFP) or TdTomato (Red). The NPCs from both species were dissociated together and mixed before injection into brains of mice at postnatal day P4-6. The brains of the animals were harvested after 1, 2, 4, 6, 8 and 19 weeks post transplantation (PST). Each brain was cryosectioned and the anatomy of GFP-

*Figure 4 continued on next page*

*Figure 4 continued*

and TdTomato-positive pyramidal neurons in the cortex was analyzed. Parameters used for the morphometric analysis are listed. (B) Representative confocal images from human and chimpanzee pyramidal neurons eight weeks PST. Arrowheads indicate potential apical dendrites. Scale bar = 10 μm. (C) Number of pyramidal neurons used for the morphometric analysis at different time points following transplantation. (D–K) Graphs illustrating changes in morphometric variables in human and chimpanzee iPSC-derived pyramidal neurons over time after transplantation. (D, E) Total dendritic length; (F, G) mean segment length; (H, I) dendritic spine number; and (J, K) dendritic spine density. Each point in the bar graphs represents an individual neuron. Statistical significance is shown between adjacent time points and between the earliest (1 week) and the latest (19 weeks) time point after transplantation. The top left corner insets show regression lines (red) of respective variables correlated with weeks PST. Coefficient of determination ($R^2$) and statistical significance are indicated for each regression analysis (***$p < 0.001$, Pearson's correlation test). (L–N) Comparative morphometric analyses on iPSC-derived transplanted neurons showed significant difference between species on specific time points. (L) Total dendritic length (TDL: 2 weeks, $p = 0.0100$; 8 weeks, $p = 0.0039$; 19 weeks, $p = 0.0316$); (M) mean segment length (MSL: 6 weeks, $p = 0.0483$; 8 weeks, $p = 0.0086$; 19 weeks, $p = 0.0016$); and (N) dendritic spine number (DSN: 8 weeks, $p = 0.0482$). (*$p < 0.05$; **$p < 0.01$, Unpaired t test).

DOI: https://doi.org/10.7554/eLife.37527.009

The following figure supplements are available for figure 4:

**Figure supplement 1.** In vivo differentiation of human and chimpanzee neurons post-transplantation.

DOI: https://doi.org/10.7554/eLife.37527.010

**Figure supplement 2.** Morphometric analysis of human and chimpanzee neurons post- transplantation.

DOI: https://doi.org/10.7554/eLife.37527.011

The variables reflecting dendritic spine number and density (DSN and DSD) in iPSC-derived neurons significantly differed across the time points examined for both species (*Figure 4H–K*). The correlation between these morphometric variables and time was calculated via regression analysis to define the coefficient of determination ($R^2$), and statistical significance was determined for each regression analysis. The regression analysis allows us to summarize and study relationships between two continuous quantitative variables (morphometric measurements and time) (insets on *Figure 4D–K* and *Figure 4—figure supplement 2A–F*). The slopes that showed significant positive correlation for both human and chimpanzee values were DSD and DSN, indicating that the number of dendritic spines from transplanted neurons increased over time, implying neuronal maturation for both species.

## Cross-species comparison revealed significant changes in timing of dendritic maturation

The absolute values and the timing of developmental events differed between the species for most of the variables examined, but the extent of the differences varied (*Figure 4L–N* and *Figure 4—figure supplement 2G–I*). Comparison between human and chimpanzee iPSC-derived neurons revealed a significant effect of species for the variables TDL, MSL, DSN and TREE at different time points.

The growth of dendrites, quantified as increase in TDL over the 19 week period, displayed different trajectories in each species (*Figure 4L*). At four weeks PST, neurites from human transplanted neurons began elongating and continued growing until the last time point examined (19 weeks). In contrast, the length of dendrites in chimpanzee iPSC-derived neurons increased at two weeks PST, with little change afterward. The difference between the species was significant at 2, 8 and 19 weeks. Similarly, MSL was significantly higher in human neurons compared to chimpanzee neurons at 6, 8 and 19 weeks (*Figure 4M*). The DSN displayed little difference between the species for the first six weeks PST but, at 8 and 19 weeks, the DSN increased rapidly in the human iPSC-derived neurons, producing significantly higher numbers of spines compared to the chimpanzee iPSC-derived neurons at eight weeks (*Figure 4N*).

Our results suggest that some aspects of dendritic morphology develop at earlier time-points in chimpanzee neurons compared to human neurons (1–2 weeks; TDL, MSL, and DSN). However, the pattern is not seen in all of the dendritic variables (*Figure 4L–N* and *Figure 4—figure supplement 2 G, H and J*), indicating that parts of dendrites with different functions may mature at different rates in each species.

## Differential functional maturation of human, chimpanzee and bonobo neurons

We then asked if there were functional changes related to the differences observed in morphological neuronal maturation. To quantify neuronal function over time, we differentiated NPC lines from four human, two chimpanzee and two bonobo individuals (total of eight individuals and two lines each) (differentiation scheme, *Figure 5A*). The cells stopped dividing and neural processes became apparent in the dish. At this stage, neurons were positive for pan-neuronal markers (Tuj1, Map2AB, Synapsin1) (*Figure 5B and C*) and cortical markers (*Figure 5—figure supplement 1*). We did not observe significant differences in the percentage of cells positive for these markers between human, chimpanzee and bonobo lines at two, four or six weeks of differentiation (*Figure 5C* and *Figure 5—figure supplement 1*). To address neuronal development functionally over time, we used the multi-well microelectrode array platform (MEA, Maestro, *Axion Biosystems, Figure 5D*). The system allows for simultaneous recordings of extracellular voltage changes associated with electrical events on the cell lines, thereby allowing for the detection of functional changes during the course of neuronal maturation (one to six weeks). We then analyzed the normalized neuronal spiking and network bursts from differentiating neurons over time. While chimpanzee and bonobo cultures showed significantly increased firing rates at earlier time points (two weeks after differentiation), human cultures showed increased firing rates at later time points (six weeks after differentiation, *Figure 5E and F*). The functional data corroborated the developmental differences observed in the morphological features between human and chimpanzee transplanted cells (*Figure 4L–M*). Interestingly, there was a similar time window (about four to five weeks) when human neuronal developmental features exceeded those of NHPs both in vitro and in vivo (compare *Figure 4L* with *Figure 5F*), indicating the presence of species-specific cell-autonomous features consistent across the two lines of analysis.

## Discussion

Here we present evidence for differential migration patterns in human NPCs compared to those in chimpanzees and bonobos and for differences in the early development of pyramidal neurons in humans and chimpanzees. We reveal that the species-specific differences in migration remain preserved after NPC grafting into the mouse cortex, suggesting a strong cell-autonomous effect. Further molecular and mechanistic analyses should address the question of which specific genes – or pathways – underlie the observed differences in migration between the two species. The observation of differential gene expression, carries implications that extend beyond evolutionary studies, since differential expression of genes related to cell migration and aberrant in vitro cell migration has been reported in NPCs derived from patients with neuropsychiatric disorders (*Brennand et al., 2015*). Our results support the notion that NPC migration is a tightly regulated process in primates, and changes in features related to this process are correlated with alterations in fetal cortical neurogenesis and, potentially, pathogenesis. Recently, Otani et al. showed that the level of individual cortical progenitor cell clonal output in primates is also regulated cell autonomously, opening the possibility for comparative in vitro studies using human and NHP NPCs (*Otani et al., 2016*). This approach will be essential to help define the neurodevelopmental pathways that shaped human uniqueness and whose disruption is likely to contribute to neural developmental disorders.

One important consideration when modeling neurodevelopment in a dish with iPSC-derived cells is to have relevant milestones recapitulated in a defined cellular type. For the CNS, not only is the neuronal subtype relevant, but extrinsic factors also play a crucial role in the neuronal maturation. To overcome this limitation and to take advantage of a neutral in vivo setting, we engrafted iPSC-derived NPCs into mouse brains, using the rodent cortex as a scaffold for neuronal development. By analyzing the change in the morphology of transplanted neurons over time, it was possible to investigate neurodevelopmental aspects of two distinct primate species exposed to the same in vivo environment (mouse cortex). Our focus was specifically on the morphology of pyramidal neurons. Although pyramidal neurons represent a morphological class of neurons with different functional properties (*Chagnac-Amitai et al., 1990*; *Connors and Gutnick, 1990*; *Mainen and Sejnowski, 1996*), we have adhered to the morphological definition of pyramidal neurons (*DeFelipe and Fariñas, 1992*; *Nieuwenhuys, 1994*; *Spruston, 2008*), which, combined with the quantification of neuronal morphology as utilized in post-mortem studies, enabled a direct comparison of our findings with previous research (*Bianchi et al., 2013a*; *Petanjek et al., 2008*; *Mrzljak et al., 1988*;

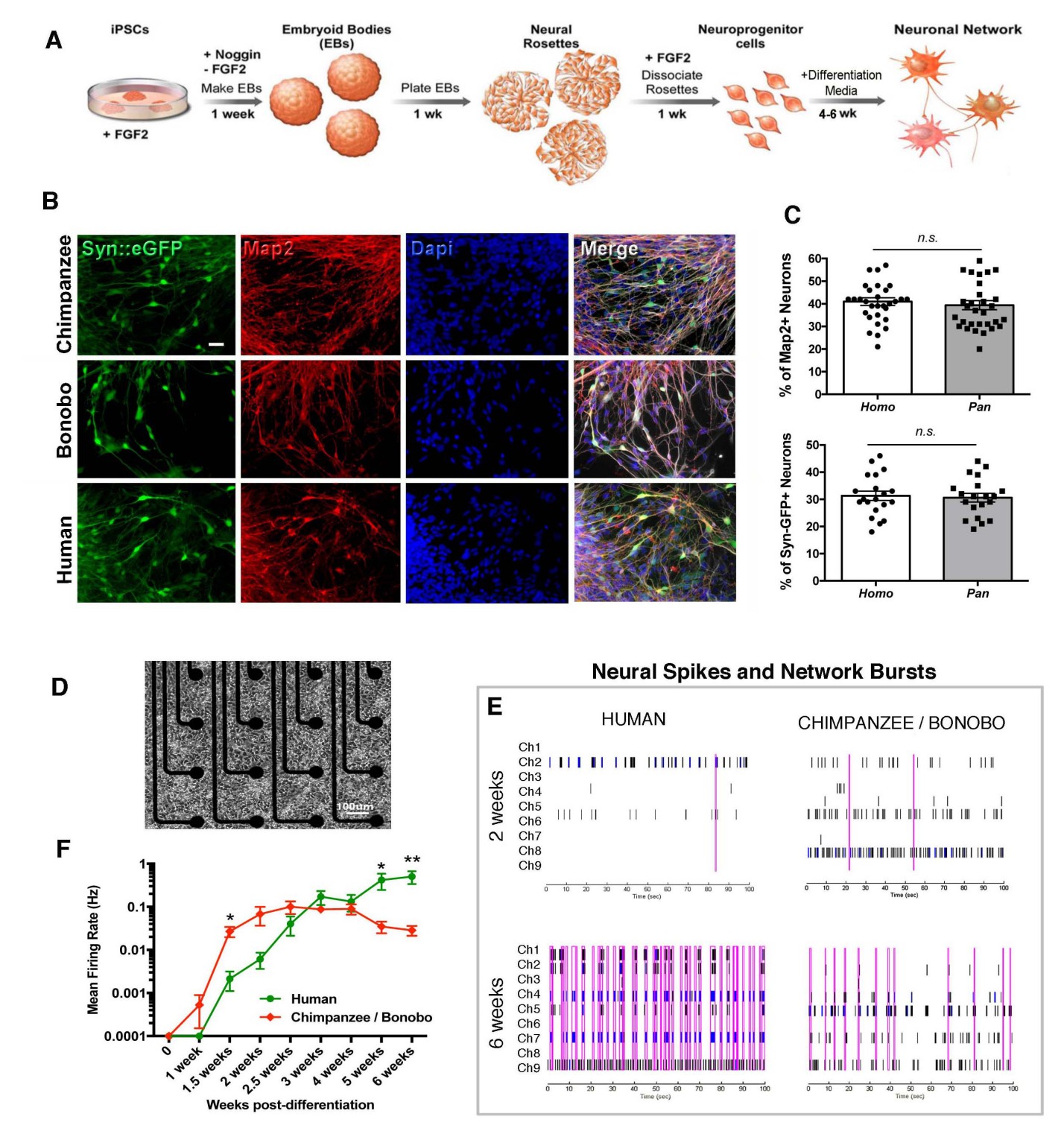

**Figure 5.** Neuronal functional analysis of human, chimpanzee and bonobo over time. (**A**) Differentiation scheme using EBs, neural rosettes and NPCs as intermediate cell lines. (**B**) iPSC-derived neurons in culture can be visualized by infection with a lentiviral vector carrying the EGFP reporter gene under the activity of the neuron-specific Synapsin promoter (Syn::eGFP). Most of the eGFP-positive neurons expressed the mature neuronal marker Synapsin. Scale bar: 20 μm. (**C**) No significant change was observed in the percentage of neurons expressing the neuronal markers Map2AB and Synapsin::eGFP between human, chimpanzee and bonobo lines. (**D–E**) Multielectrode array (MEA) analysis was performed in differentiating neurons from human, chimpanzee and bonobo lines. (**D**) Example of chimpanzee neurons growing on top of electrodes on a multiwell plate. (**E**) Representative MEA traces

*Figure 5 continued on next page*

*Figure 5 continued*

on two- and six-week-old neurons from humans, chimpanzees and bonobos showing neural spikes (black dashes), neural bursts (blue dashes) and network bursts among channels (pink lines). (F) Comparative analysis between species over differentiation time showing that while chimpanzee and bonobo neurons have accelerated functional maturation (1.5 weeks, p=0.0147), human neural cultures show increased firing rates at later time points (5 weeks, p=0.0239 and 6 weeks, p=0.0018). (*p<0.05; **p<0.01, Unpaired t test).

DOI: https://doi.org/10.7554/eLife.37527.012

The following figure supplement is available for figure 5:

**Figure supplement 1.** Immunocytochemistry of iPSC-derived neurons from humans and chimpanzees.

DOI: https://doi.org/10.7554/eLife.37527.013

*Mrzljak et al., 1992*). Our findings suggest that transplanted human and chimpanzee progenitors can differentiate into neurons, with the morphology typical of pyramidal cells, and can display patterns of development similar to those documented in the literature (*Mrzljak et al., 1988*). At the same time, iPSC-derived neurons from the two species can also display differences in the timing of certain developmental events, suggesting the role of cell-autonomous developmental programs typical of each species.

The differences in neuronal development between *Homo* and *Pan* are often examined in the context of changes in timing or rate (i.e., heterochronic changes) of specific developmental events (*Vrba, 1998*; *McKinney, 2002*; *Leigh, 2004*). The present study supports this approach by identifying three distinct processes – NPC migration, dendritic development, and functional maturation of neurons – that also display differences in timing of their development in humans compared to the *Pan* species analyzed. Previous reports comparing human, chimpanzee and macaque post-mortem brain expression profiles reported that the most prominent human-specific expression changes during development affected genes associated with synaptic functions, representing an extreme shift in the timing of synaptic development in the human prefrontal cortex (*Liu et al., 2012*; *Somel et al., 2009*). A more specific focus on one developmental parameter, coupled with gene expression studies, could reveal a more consistent pattern, especially if applied to a larger number of species. In addition to their evolutionary importance for specifying species-specific features, early developmental events are important for understanding the neuroanatomical changes observed in neurodevelopmental disorders, since pathologies in dendritic organization tend to appear during the initial establishment of dendrites in Down syndrome, Rett syndrome, autism spectrum disorder, and Fragile X (*Marin-Padilla, 1972*; *Belichenko et al., 1994*; *Hutsler and Zhang, 2010*; *Irwin et al., 2001*; *Liu et al., 2016*).

Thus, in the present study we have observed several patterns that are largely in accordance with heterochrony, a mechanism previously suggested to underlie changes in human brain size and organization during its evolutionary history (*Vrba, 1998*; *McKinney, 2002*; *Leigh, 2004*). Focusing specifically on the morphology of dendrites, post-mortem studies have suggested that the dendrites of human pyramidal neurons in adulthood are longer and more branched and have a higher number of dendritic spines compared to the same class of neurons in the chimpanzee neocortex (*Bianchi et al., 2013b*). The analysis of dendritic development of pyramidal neurons in the two species using iPSC-derived neurons suggested that the observed difference in adulthood might be a consequence of differences in the early development between humans and chimpanzees. As our results suggest, dendrites on human neurons are initially shorter and less branched and have fewer dendritic spines compared to the dendrites of chimpanzee iPSC-derived neurons. At one point in development – ranging from six weeks for MSL and eight weeks for TDL and DSN – dendrites on human iPSC-derived neurons enter a rapid growth and surpass the chimpanzee iPSC-derived neurons (*Figure 4L–N*), suggesting the possibility of late initiation of dendritic growth in humans.

The morphological findings are supported by the analysis of neuronal function. The human neurons in culture also displayed functional developmental delay compared to chimpanzee and bonobo neurons, with human neurons showing increased firing rates and neuronal bursts at later time points. Interestingly, we observed a similar time window (between four and five weeks) wherein human neuronal developmental parameters surpass NHPs in both in vivo and in vitro analyses (compare *Figure 4L and N* with *Figure 5F*). Our results suggest the presence of cell-autonomous characteristics of neuronal maturation in humans that are different from those of chimpanzees and bonobos and support the idea that developmental delay in humans results in increased neuronal complexity

(increased dendritic length, spine density) and neuronal function (increased firing rates and network bursts) at later time points during maturation.

As soft tissue remains scarce in human developmental and evolutionary neuroanatomy studies, the novel strategy we present here - combining the use of iPSC technology with in vitro analysis of cell migration and in vivo grafting - offers a paradigm aimed at elucidating the genetic underpinnings of the observed changes and the evolutionary timing of human emergence and suggests further implications for the evolution of cortical microcircuitry across different primate clades.

## Materials and methods

### Cell line statement

Fibroblasts from three human subjects (WT126, WT33, Adrc40), two chimpanzees (PR00818 and PR01209), two bonobos (PR01086 and AG05253), were reprogrammed to induced pluripotent stem cells (iPSC) at the Salk Institute for Biological Studies (La Jolla, CA) and differentiated into neural progenitor cell as previously described (*Marchetto et al., 2013*; *Marchetto et al., 2010*). Fibroblasts from chimpanzee and bonobo were obtained from Coriell Institute and their use was approved by the US Fish and Wildlife Service, under permit number MA206206. Protocols describing the use of human iPSCs were previously approved by the University of California, San Diego (UCSD) and the Salk Institute Institutional Review Board. Authentication: Cell identity was confirmed by G-banding karyotyping and routine immunofluorescence assays. Primary antibodies used in this study to characterize the cell lines were: human Nestin (1:250, Chemicon cat# ABD69) and SOX2 (1:200, Santa Cruz cat# sc-17320), for neuroprogenitor cells, and Map2 (1:500, Abcam, cat# ab5392); GFP (1:500, Invitrogen, cat# A6455); and CTIP2 (1:200, Abcam, cat# ab18465), for neurons. Mycoplasma Contamination: All the lines have been tested negative for Mycoplasma contamination. We test all lines on a monthly basis using a mycoplasma detection kit that uses polymerase chain reaction (PCR) and can detect over 95% of commonly occurring mycoplasma species contaminating cell lines.

### Cell culture reprogramming and neuronal differentiation

Fibroblasts from humans and nonhuman primates were reprogrammed to iPSCs using Yamanaka retroviral vectors expressing reprograming genes (Klf4, Oct4, Sox2 and cMyc) and differentiated into NPCs as previously described (*Marchetto et al., 2013*; *Marchetto et al., 2010*). Established iPSC colonies were kept in feeder-free conditions indefinitely and passed using mechanical dissociation. To obtain NPCs from iPSCs, EBs were formed by mechanical dissociation of iPSC clusters and plated into low-adherence dishes in DMEM/F12 plus N2 and B27 (Invitrogen) medium in the presence of Noggin (R and D) for forebrain induction for approximately seven days. Then, floating EBs were plated onto poly-ornithine/laminin (Sigma)-coated dishes in DMEM/F12 supplemented with N2 and B27 (Invitrogen) and Noggin. Rosettes were collected after seven days. Rosettes were then dissociated with Accutase (Chemicon) and plated again onto coated dishes in DMEM/F12 supplemented with N2 and B27 and 10 ng/ml of FGF2 (R and D). Homogeneous populations of NPCs were obtained after one to two passages with Accutase under the same conditions. To obtain neurons, NPCs were cultured in DMEM/F12 supplemented with N2 and B27, laminin (1 µg/ml), BDNF (20 ng/ml), GDNF (20 ng/ml) and cyclic AMP (500 µg/ml) for up to eight weeks.

### Immunofluorescence in vitro

Cells were briefly fixed in 4% paraformaldehyde for 10 min and then permeabilized with 0.5% Triton-X100 in PBS. Cells were then blocked in 0.5% Triton-X100 with 5% donkey serum for 1 hr before incubation with primary antibody overnight at 4°C. After three washes in PBS, cells were incubated with secondary antibodies (Jackson ImmunoResearch) for 1 hr at room temperature. Fluorescent signals were detected using a Zeiss LSM 710 Laser Scanning Confocal Microscope and images were processed with Photoshop CS3 (Adobe Systems). Primary antibodies used in this study were human Nestin (1:100, Chemicon); Tuj-1 (1:500, Covance); Map2 (1:100, Sigma); GFP (1:200, Molecular Probes-Invitrogen) Synapsin1 (1:200, Calbiochem), GABA (1:200, Sigma) and CTIP2 (1:200, Abcam).

## Single cell capture and library preparation for RNA sequencing

NPCs were dissociated into single cell suspension with Accutase enzyme (Chemicon) and cell suspension from each sample (human [wt33 c1], chimpanzee [PR00818 c5] and bonobo [PR01086 c1]) were directly loaded on each sample well to generate single-cell GEMs (gel beads in emersion) using Chromium Single Cell Controller (10X Genomics Inc). Approximately 10,000–13,000 NPCs were captured and single-cell RNA-Seq libraries were prepared using Single-Cell 3′ Gel Bead and Library Kit (10X Genomics Inc) following manufacturer's instruction. The barcoded sequencing libraries were loaded and sequenced on Illumina NextSeq 500 sequencer using 98 bp for Read1, 8 bp for I7 Index, 26 bp for Read2 with an average sequencing depth of 12,000–13,000 reads per cell.

## Single cell transcriptome analysis

Sequenced samples were processed using the Cell Ranger 1.2 pipeline (10X Genomics Inc) with an expectation of 7,000 cells per sample. All reads were mapped to the GRCh38-1.2.0 reference assembly. Analysis of reads mapped to transcriptome showed an average of 2306 genes/cell for human [wt33 c1], 2606 genes/cell for chimpanzee [PR00818 c5] and 2756 genes/cell for bonobo [PR01086 c1]. To reduce the dimensionality of the cell-by-gene expression matrix and visualize the diversity of gene expression among NPCs of different primate species in a 2-dimensional scatter plot, we applied the t-distributed stochastic neighborhood-embedding (tSNE) algorithm (*Mahfouz et al., 2015*). The default parameters from Cell Ranger were used for tSNE (perplexity = 30, theta = 0.5). To determine whether there was a difference in the cortical layer identity between species, we referred to *Molyneaux et al. (2007)* for markers for each cortical layer (*Molyneaux et al., 2007*). Region membership was calculated by summing the presence or absence of marker gene expression within the SOX2 positive population. Layer Vb = FOXO1, ECPN, LIX1, SYT9, S100A10, OMA1, LDB2, CRIM1, PCP4, RAC3, and DIAP3; layer VI = LXN, FOXP2, DARPP32, IGH6; layer II/III = RORB, CYP39A1, LHX2, UNC5D, GPR6, MEF2C, DTX4, CUX1, CUX2, KITL, SVETL; Hindbrain = EN2, PAX2, EN1, ISL1, LHX3. Each species showed similar proportions of layer markers, indicating that within the cortex there was no particular preference for a given layer between the species (*Figure 1—figure supplement 2A*).

## RNA sequencing analysis of NPCs

Total cellular RNA was extracted from ~$5 \times 10^6$ cells using the RNeasy Protect Mini kit or RNeasy Plus kit (Qiagen, Valencia, CA), according to the manufacturer's instructions. RNA quality was assayed using Agilent Technologies 2200 TapeStation and samples with integrity superior to RIN 8.5 were used for library preparation. PolyA+ RNA was fragmented and prepared into sequencing libraries using the Illumina TruSeq RNA sample preparation kit and analyzed on an Illumina HiSeq 2000 sequencer at the UCSD Biomedical Genomics Laboratory (BIOGEM). Libraries were sequenced using single-end 100 bp reads at a depth of 15–30 million reads per library (100 ± 25 bp fragments). RNA-sequencing FASTA files were aligned to the Human GRCh37 reference and separately to the *Pan troglodytes* CSAC 2.1.4 reference using the STAR aligner (*Dobin et al., 2013*). All reads from human, chimpanzee and bonobo species were filtered to retain only those that uniquely mapped to both the human and the *Pan troglodytes* reference sequence, with a maximum of three mismatches to the species of origin and a maximum of five mismatches to the alternate species. There was no noticeable decrease in alignment or expression for the bonobo sample. The filtered set of reads was annotated using the human genome coordinates and the respective Ensembl gene annotations. Counts per gene were calculated using the HTseq-count algorithm (*Anders et al., 2015*). Differential expression was performed using EdgeR (*Robinson et al., 2010*) and a threshold of adjusted p value (padj) <0.05 was used to call differentially expressed genes. GO term annotation was performed using BioMart available from Bioconductor (*Smedley et al., 2015*). For the heatmap display, expression was first normalized by sample to transcripts per million (TPM) and then normalized by gene using a Z-score calculation. Principal components analysis was performed on all genes and samples using the R pca Methods package. RNA-seq data have been deposited in the Gene Expression Omnibus.

## Scratch assay and live imaging of NPC migration

Human, chimpanzee or bonobo NPCs were transfected with a GFP-expressing plasmid (pmaxGFP Vector) using the Nucleofactor technology as described by the manufacturer (Transfection kit VPG-1005, Lonza, Basel, Switzerland) and plated on 3 mm coverslip bottom dishes (GWst-3533, WilCO Wells BV). Once cells reached confluence, six scratches (three parallel to each other and vertical to other three) were made using a glass pipette (*Figure 2—figure supplement 1A*). Several regions of interest along the scratches were imaged every 10 min for varying times (often for 24 hr or 48 hr), taking 5 z-sections using a Pln Apo 20 × 0.8 NA objective on an inverted Zeiss CSU Spinning Disk Confocal Microscope (Zeiss). Cells were kept at 37°C and under 5% $CO_2$ during imaging in a stage-top incubation system (Zeiss).

The table below summarizes the number of movies recorded and cell lines analyzed per species.

| Species | Number of Cell Lines (NPCs) | Number of clones for each cell line | Number of movies | Number of cells analyzed |
|---|---|---|---|---|
| Human | 4 (WT33, WT126, A40, H9) | 2 each + 1 (H9) | 57 | 340 |
| Chimpanzee | 2 (PR00818, PR01209) | 2 each | 36 | 165 |
| Bonobo | 2 (AG05253B, PR01086) | 2 each | 34 | 186 |

## Tracking and analyzing migrating NPCs

Optical z-sections were flattened and color levels were adjusted to make single cells distinguishable from each other on ImageJ (NIH). Individual cells were tracked using MTrackJ plugin (*Meijering et al., 2012*) by personnel who were blind to the working hypothesis (*Figure 2—figure supplement 1*). The health of the cells and scratch closure were monitored using the DIC channel. To confirm that the filling-in process after scratch assay was due to NPCs migrating toward the scratch rather than simply a random walk process we calculated two features for each NPC: 1) cumulative distance travelled (CDT) and 2) distance difference (DD) metric. NPCs with a CDT shorter than a quarter of the viewing area were not included in the calculations to prevent bias from non-migrating cells. The DD metric measures the net displacement (path independent) either toward (0 to positive infinity, μm) or away (0 to negative infinity, μm) from the scratch. All the migration paths were plotted with respect to the center of the scratch, denoted with a gray line. Using this criterion, we found that 84% of human cells moved toward the scratch (*Figure 2—figure supplement 1B*, black line), 76% of chimpanzee cells moved toward the scratch (*Figure 2—figure supplement 1C*, red line), and 72% of bonobo cells moved toward the scratch (*Figure 2—figure supplement 1D*, blue line). To test the significance of the DD toward the scratch for each species' NPCs, we simulated random motion paths with random scratch locations (N = 200 random cells/species). For each species, the distribution of random DD was uniform around 0 (*Figure 2—figure supplement 1B–D*), consistent with a random migration process that would have no bias either toward or away from the scratch.

## Calculation of migration speed

Migration speed was calculated in two ways. First, the mean speed was determined as the mean of the distance traveled over the time for each tracked cell. We also confirmed this calculation by determining the cumulative distance traveled (CDT) and then fit this to a linear regression with the slope serving as the mean speed (distance covered/time), which yielded equivalent values for the speed of migration.

## Neurosphere migration assay

The neurosphere migration assay described here was performed according to previously published protocols (*Brennand et al., 2015*; *Delaloy et al., 2010*). Briefly, NPCs were dissociated with Accutase and then cultured for 72 hr in non-adherent plates to generate neurospheres. Neurosphere culture is defined as a culture system composed of free-floating clusters of NPCs. Neurospheres were then manually picked and plated in Matrigel matrix (0.5 mg Matrigel was used to coat a 24-well

plate 1 hr before neurosphere plating) and fixed 48 hr later to access NPC migration. Average NPC migration distance from each neurosphere was measured using Image J software (NIH).

## NPC preparation and grafting

Transplantation was performed using human or chimpanzee NPCs that had previously been infected with lentiviral vector constitutively expressing EGFP (enhanced green fluorescent protein) or TdTom (tandem dimer tomato) and had been passaged for at least three passages before grafting to eliminate lentiviral particles from the culture media prior to grafting. NPCs were lifted from the culture plate using Accutase enzyme and re-suspended in saline buffer (PBS) with 0.5% BSA, ROK inhibitor, BDNF, ascorbic acid, cyclic AMP, and laminin, at 50,000 cells per µl. On postnatal day 3–5 NOD-SCID (Nonobese diabetic/severe combined immunodeficiency), mouse pups were anesthetized on ice and 0.5 µl of cell suspension was delivered to the right hemisphere of the frontal motor cortex by stereotaxic surgery. We chose motor cortex as the grafting site because it represents an area that is similar in rodents and primates both in organization and in function. The injection site was determined using the difference between bregma and lamba (*d*), using the position of the bregma as reference: anterior-posterior -(1/2) x *d* mm; lateral, 1 mm; ventral, 0.5 mm (from dura). To observe in vivo migration, NPCs from two human and two chimpanzee lines were grafted into the developing postnatal mouse cortex. Mice were sacrificed at two weeks PST and two different migration parameters were analyzed: 1) area covered by migrating NPCs and 2) distance migrated by NPCs from injection site (*Figure 3*). To observe the temporal sequence of neuronal development, the mice were sacrificed at 1, 2, 4, 6, 8 and 19 weeks post-transplantation and the morphology of the transplanted cells differentiated into pyramidal neurons was quantified at each of the time points (see scheme on *Figure 4A*). We chose to focus specifically on pyramidal neurons since they are easy to identify based on their morphology, they represent the most common type of neuron in the cortex, and differences in dendritic organization have previously been reported between the two species under study, namely humans and the common chimpanzee (*Bianchi et al., 2013a*). The time period under study spans the time points at which, based on post-mortem studies in humans, pyramidal neurons in the cortex commence their post-natal development and accomplish most of their growth (*Marin-Padilla, 1970*). All experimental procedures were approved by the Institutional Animal Care and Use Committee at The Salk Institute for Biological Studies.

## Tissue preparation

At 1, 2, 4, 6, 8 and 19 weeks PST, animals were anesthetized with ketamine/xylazine (100 mg/kg and 10 mg/kg, respectively) and perfused transcardially with 0.9% saline followed by 4% paraformaldehyde. The brain samples were dissected and post-fixed with 4% paraformaldehyde for 18 hr and equilibrated in 30% sucrose at 4°C. Coronal sections of 40 µm thickness were prepared for samples from one-, two- and four-week PST brains. As the dendrites became too large to be contained within 40 µm sections, samples from 6-, 8- and 19 week PST brains were prepared at 80 µm using a sliding microtome. Tissue was mounted onto glass microscope sides with glycerol mounting medium with DAPI. The sections were immunostained with the antibodies GFP (1:400, Molecular probes) and TdTomato (1:400, LifeSpan Biosciences, Inc) to increase the signal of human and chimpanzee transplanted neurons. CTIP2 antibody (1:200, Abcam) and SATB2 (1:200, Abcam) were used to characterize cortical layer identity of the transplanted cells (*Figure 4—figure supplement 1A–D*).

## Quantification of neuronal morphology

The analysis of neuronal morphology was conducted on transplanted cells located in the cortex and within the callosal white matter. Only the neurons that were relatively isolated from other neurons or neuronal clusters and had a fluorescent signal strong enough to allow for accurate tracings were included in the analysis. In addition, the analyzed neurons had to display distinct pyramidal-shaped somata co-stained with DAPI, at least one dendrite emerging from the soma, and dendritic trees that were free from obvious signs of cutting- or other processing-related damages. Dendritic morphology was quantified along x, y, and z coordinates using 'Live Image' option on Neurolucida v.10 software (MBF Bioscience, Williston, VT) interfaced with Nikon Eclipse E600 using a Plan Fluor 40x/ 1.30 oil lens. In accordance with other studies in humans (*Petanjek et al., 2008*; *Jacobs et al., 2001*) and chimpanzees (*Bianchi et al., 2013a*; *Bianchi et al., 2013b*), we conducted dendritic tree analysis

with the following variables: soma size (SOMA), the surface area of the cell body; total dendritic length (TDL), the summed length of all dendrites/neuron; tree number (TREE), the total number of dendritic trees/neuron; dendritic segment count (DSC), the number of dendritic segments per neuron; dendritic spine number (DSN), the total number of dendritic spines/neuron; and dendritic spine density (DSD), the average number of spines/1 μm of dendritic length. Dendritic segments were assigned numbers following a centrifugal ordering scheme. The segments closest to the soma were termed first order segments; those after the first branching point were second order segments, with segment order increasing after each branching point. Neuronal morphology was quantified from a total of 141 neurons, encompassing 75 chimpanzee and 66 human cells, that is between 6 and 18 cells per species for each time point (*Figure 4C*).

### Confocal imaging

Fixed, unstained brain sections were imaged using confocal microscopy. Images were acquired with the 63x or 100x objectives on a Zeiss LSM 710 Laser Scanning Confocal Microscope. Confocal microscopy images were taken from representative images of neurons after morphometric analysis was performed.

### Multielectrode array analysis (MEA)

Using the 96-well MEA plate from Axion Biosystems, we plated cells derived from four humans, two chimpanzees and two bonobos (two clones from each line) and we repeated the experiment three times. Each well was seeded with 15,000 NPCs that were induced into neuronal differentiation as described previously and summarized on *Figure 5A*. Each well was coated with poly-ornithine and laminin prior to cell seeding. Cells were fed three times a week and measurements were taken before media changes. Recordings were performed in a Maestro MEA system and AxIS software (Axion Biosystems) using a bandwidth with a filter for 10 Hz to 2.5 kHz cutoff frequencies. Spike detection was performed using an adaptive threshold set to 5.5 times the standard deviation of the estimated noise on each electrode. Each plate rested for 10 min for acclimatization in the Maestro instrument and then was recorded for an additional 10 min to calculate spike and burst rates. Multi-electrode data analysis was performed using the Axion Biosystems Neural Metrics Tool, and an active electrode was considered once five spikes occurred over the period of 1 min (five spikes/min). Bursts were identified in the data recorded from each individual electrode using an adaptive Poisson surprise algorithm. Network bursts were identified for each well, using a non-adaptive algorithm requiring a minimum of 10 spikes with a maximum inter-spike interval of 100 ms.

### Statistical analysis

Statistical analysis was conducted using PRISM software (Graph Pad, San Diego, CA). Morphometric and neuronal functional variables between the species were examined with unpaired t-tests for each time point. Where appropriate, ANOVA, Mann-Whitney U and Student's t-test post-hoc analysis was performed with Bonferroni correction for multiple comparisons, and the p-values presented are those of the corrected values. Linear regression analysis was performed for respective variables (morphometric measurements) correlated with weeks (time) post transplantation. Statistical significance was determined with Pearson's correlation test.

### Data availability

Source for bulk and single cell RNA-seq data sets are available with GEO accession code GSE124706.

## Acknowledgments

This work was supported by the National Institutes of Health (NIH) grants MH095741, MH088485, DP2-OD006495-01, R01MH094753, R01MH103134, R01MH100175, K99MH101634, and U19MH107367; the California Institute for Regenerative Medicine (CIRM) TR2-01814 and TR4-06747; a NARSAD Independent Investigator award, The G Harold and Leila Y Mathers Foundation, The Leona M. and Harry B Helmsley Charitable Trust, and JPB Foundation. KP is supported by NIH MH101634, MH113924, a Crick-Jacobs Junior Fellowship, a NARSAD Young Investigator Award,

and the Lieber Foundation. This work was supported by the Next Generation Sequencing Core (NGS) Facility of the Salk Institute with funding from NIH-NCI CCSG: P30 014195, the Chapman Foundation and the Helmsley Charitable Trust. The authors would like to thank M.L. Gage for editorial comments.

## Additional information

### Competing interests

Fred H Gage: Reviewing editor, *eLife*. Alysson R Muotri: is a co-founder and has equity interest in TISMOO, a company dedicated to genetic analysis and cellular models focusing on therapeutic applications customized for autism spectrum disorder and other neurological disorders with genetic origins. The terms of this arrangement have been reviewed and approved by the University of California San Diego in accordance with its conflict of interest policies. The other authors declare that no competing interests exist.

### Funding

| Funder | Grant reference number | Author |
|---|---|---|
| National Institutes of Health | Research Project Grant | Maria C Marchetto<br>Alysson R Muotri<br>Krishnan Padmanabhan<br>Fred H Gage |
| California Institute for Regenerative Medicine | Research Grant | Alysson R Muotri |
| National Institutes of Health | Pathway to independence award | Krishnan Padmanabhan |
| National Alliance for Research on Schizophrenia and Depression | Young Investigator award | Krishnan Padmanabhan |
| Leona M. and Harry B. Helmsley Charitable Trust | Research Project Grant | Fred H Gage |
| Salk Cancer Center | NCI P30 CA014195 | Fred H Gage |

The funders had no role in study design, data collection and interpretation, or the decision to submit the work for publication.

### Author contributions

Maria C Marchetto, Conceptualization, Formal analysis, Supervision, Funding acquisition, Visualization, Methodology, Writing—original draft, Project administration, Writing—review and editing, Revising the manuscript critically for important intellectual content; Branka Hrvoj-Mihic, Conceptualization, Supervision, Validation, Investigation, Methodology, Writing—original draft, Writing—review and editing; Bilal E Kerman, Conceptualization, Formal analysis, Supervision, Investigation, Methodology, Writing—original draft, Writing—review and editing; Diana X Yu, Conceptualization, Formal analysis, Investigation, Visualization, Methodology, Writing—original draft; Krishna C Vadodaria, Formal analysis, Supervision, Investigation, Visualization, Methodology, Writing—original draft, Writing—review and editing; Sara B Linker, Data curation, Formal analysis, Visualization, Methodology, Writing—review and editing; Iñigo Narvaiza, Conceptualization, Formal analysis, Investigation, Methodology; Renata Santos, Ruth Oefner, Investigation, Visualization, Methodology; Ahmet M Denli, Formal analysis, Methodology, Writing—review and editing; Ana PD Mendes, Investigation, Visualization, Methodology, Writing—review and editing; Jonathan Cook, Formal analysis, Investigation, Methodology; Lauren McHenry, Formal analysis, Methodology, Writing—original draft; Jaeson M Grasmick, Kelly Heard, Callie Fredlender, Rea Xenitopoulos, Formal analysis, Visualization, Methodology; Lynne Randolph-Moore, Grace Chou, Data curation, Formal analysis, Methodology; Rijul Kshirsagar, Formal analysis, Methodology; Nasun Hah, Resources, Formal analysis, Writing—review and editing, Experimental design; Alysson R Muotri, Conceptualization, Revising the manuscript

critically for important intellectual content; Krishnan Padmanabhan, Data curation, Formal analysis, Investigation, Methodology; Katerina Semendeferi, Conceptualization, Methodology, Writing—original draft, Writing—review and editing, Interpretation, Revising the manuscript critically for important intellectual content; Fred H Gage, Conceptualization, Resources, Funding acquisition, Writing—original draft, Writing—review and editing, Revising the manuscript critically for important intellectual content

### Author ORCIDs
Maria C Marchetto (ID) http://orcid.org/0000-0002-0449-9051
Bilal E Kerman (ID) http://orcid.org/0000-0003-1106-3288
Renata Santos (ID) http://orcid.org/0000-0002-3085-5128
Fred H Gage (ID) https://orcid.org/0000-0002-0938-4106

### Ethics
Clinical trial registration
Human subjects: Fibroblasts from 3 human subjects (WT126, WT33, Adrc40), 2 chimpanzees (PR00818 and PR01209), 2 bonobo (PR01086 and AG05253), were reprogrammed to induced pluripotent stem cells (iPSC) at the Salk Institute for Biological Studies (La Jolla, CA) and differentiated into neural progenitor cell as previously described {Marchetto, 2013; Marchetto, 2010}. Fibroblasts from chimpanzee and bonobo were obtained from Coriell Institute and their use was approved by the US Fish and Wildlife Service, under permit number MA206206. Protocols describing the use of human iPSCs were previously approved by the University of California, San Diego (UCSD) and the Salk Institute Institutional Review Board. Authentication: Cell identity was confirmed by G-banding karyptyping and routine immunofloerescence assays. Primary antibodies used in this study to characterize the cell lines were: human Nestin (1:250, Chemicon cat# ABD69) and SOX2 (1:200, Santa Cruz cat# sc-17320), for neuroprogenitor cells, and Map2 (1:500, Abcam, cat# ab5392); GFP (1:500, Invitrogen, cat# A6455); and CTIP2 (1:200, Abcam, cat# ab18465), for neurons. Mycoplasma Contamination: All the lines have been tested negative for Mycoplasma contamination. We test all lines on a monthly basis using a mycoplasma detection kit that uses polymerase chain reaction (PCR) and can detect over 95% of commonly occurring mycoplasma species contaminating cell lines.

### Decision letter and Author response
Decision letter https://doi.org/10.7554/eLife.37527.023
Author response https://doi.org/10.7554/eLife.37527.024

## Additional files
### Supplementary files
• Supplementary file 1. Table showing the full list of genes differentially expressed in nonhuman primates versus human.
DOI: https://doi.org/10.7554/eLife.37527.015

• Supplementary file 2. Table showing the list of genes in the Gene Ontology (GO) category of *migration* that are differentially expressed in nonhuman primates versus human.
DOI: https://doi.org/10.7554/eLife.37527.016

• Supplementary file 3. Summary of the subjects and clones utilized for each experiment and additional information on RNA sequencing (total counts, raw counts and normalized counts).
DOI: https://doi.org/10.7554/eLife.37527.017

• Transparent reporting form
DOI: https://doi.org/10.7554/eLife.37527.018

### Data availability
RNA sequencing data comparing human samples with non human primate samples have been deposited in GEO.

The following dataset was generated:

| Author(s) | Year | Dataset title | Dataset URL | Database and Identifier |
|-----------|------|---------------|-------------|-------------------------|
| Linker SB, Gage FH, Maria C Marchetto | 2019 | Species-specific maturation profiles of human, chimpanzee and bonobo neural cells | https://www.ncbi.nlm.nih.gov/geo/query/acc.cgi?acc=GSE124706 | NCBI Gene Expression Omnibus, GSE124706 |

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
