## [Decision Letter]

Thank you for sending your article entitled "Species-specific maturation profiles of human, chimpanzee and bonobo neural cells" for peer review at *eLife*. Your article is being evaluated by three peer reviewers, and the evaluation is being overseen by a Reviewing Editor and Sabine Kastner as the Senior Editor.

Given the list of essential revisions, including new experiments, the editors and reviewers invite you to respond within the next two weeks with an action plan and timetable for the completion of the additional work. We plan to share your responses with the reviewers and then issue a binding recommendation.

General assessment:

This is an interesting manuscript reporting on a comparative analysis of developmental milestones between human, chimpanzee and bonobo cortical-like neurons. The authors use iPSCs-drived neural progenitors cells (produced in vitro) to investigate aspects of neuronal migration and differentiations among these three species. They investigate these developmental processes both in vitro and upon transplantation of iPSC-NPCs in the developing mouse cerebral cortex, followed by analysis in vivo. The work illustrates a valid and novel approach for understanding aspects of human brain development that could not otherwise be studied. In this regard, the study will be informative to the field for understanding the evolution of the human brain, and, possibly its disease.

The central conclusion of the study is that chimpanzee and bonobo neurons display similar properties as it relates to neuronal migration and maturation (dendritic tree complexity, synapse numbers and timing of synaptogenesis) and differ from human NPCs (and neurons) in these developmental parameters.

Required revisions:

Below are main points of revision that reflect suggestions that were made and agreed upon by all three reviewers.

1) There are many protocols for generating dorsalized (cortical-like) progenitors. It is generally accepted that none produce homogenous populations of NPCs of only one type. It would be important to have some further characterization, ideally at the single cell level (i.e. by scRNA-seq), of the starting NPC population (sampled at the stage soon before transplantation) to clarify any differences in the composition of the transplanted pool of cells. Related to this, some iPS lines have intrinsic fate biases, and different densities of the cultures can lead to alternative fates, so a cortical fate must be validated and quantified. The Nestin and Tuj1 analysis simply confirms the presence of neural stem cells and neurons in culture.

2) Looking at the differentially expressed (DE) genes, numerous hits relate to hindbrain identity, suggesting significant biases during in vitro differentiation. As a result, the authors may be identifying forebrain versus non-forebrain cells. In line with this, Figure 1C-E show differentially expressed genes between human and pan. However, browsing the table of DE genes suggests a potentially problematic purity difference. 14 HOX genes are among the most significant DE genes and all show the same direction change (apparently up in pan according to the supplementary file). These genes are not expressed in the mammalian cortex, suggesting that the pan lines have a differentiation bias towards hindbrain or spinal cord fates. Without access to the full expression table, it is difficult to comment on how pervasive this issue is across all individuals versus a subset. However, this result raises a flag about the purity of the cells that needs to be addressed in order to understand whether the observed enriched gene categories relate to species differences or differences in cell purity.

3) The authors should make clear how many individuals were used for each experiment, and when statistical analyses are performed on technical replicates. This must be articulated explicitly, ideally with a table of individuals and lines that went into each experiment.

4) For the migration assay in vivo, it would be useful to have low magnification images of multiple sections to get a sense for the general distribution of cells within the section, relative to the site of injection and across animals.

5) The migration assays in Figure 2 and the microelectrode array assays do not have a control for the purity of cells entering the assay. Cells from different regions may have different migratory properties and functional maturation rates that could influence the observed phenotypes. Post-staining representative sections showing that comparable populations were analyzed would help alleviate these concerns.

6) In their transplantation experiments, the authors don't provide enough insights into the types of cells that are compared. The only evidence shown is immunohistochemistry for CTIP2, but that marker only identifies a subset of cortical neurons, and may be more broadly expressed beyond cerebral cortex. Further characterization of these cells post transplantation would solve this problem.

7) For the analysis of dendritic morphology, 3D reconstructions of dendritic arbors traced in their entirety in a volume of tissue would be very important. Adding images from multiple neurons would give the reader a better sense for the differences observed and their magnitude.

---

## [Author Response]

[Editors' note: the authors’ plan for revisions was approved and the authors made a formal revised submission.]

1) There are many protocols for generating dorsalized (cortical-like) progenitors. It is generally accepted that none produce homogenous populations of NPCs of only one type. It would be important to have some further characterization, ideally at the single cell level (i.e. by scRNA-seq), of the starting NPC population (sampled at the stage soon before transplantation) to clarify any differences in the composition of the transplanted pool of cells. Related to this, some iPS lines have intrinsic fate biases, and different densities of the cultures can lead to alternative fates, so a cortical fate must be validated and quantified. The Nestin and Tuj1 analysis simply confirms the presence of neural stem cells and neurons in culture.

We performed further characterization on the starting neural progenitor population to assess the differences in composition of the different iPSC lines and to quantify other markers of cortical fate to address the purity of the progenitor pools. Specifically, we performed immunostaining to detect cortical protein expression on 4 human, 2 chimpanzee and 2 bonobo samples (new Figure 1—figure supplement 1A-C). We calculated spatial enrichment of genes in *Homo* or *Pan* NPCs using *brainImageR* tool (Linker et al., 2018) and showed that genes upregulated in human NPCs exhibited similar regional enrichment to genes upregulated in *Pan* NPCs, indicating that the differences observed in bulk RNA sequencing are not influencing broad fate specification between NPCs from *Homo* versus *Pan* (Figure 1—figure supplement 1D). We also performed single cell RNA sequencing on 1 human, 1 chimpanzee and 1 bonobo NPC line. The results from the new experiments revealed that each species had similar proportions of cortical markers, indicating that within the cortical-fated cells there was no particular preference for a given layer between the species.

We are providing the single cell RNA sequence data analysis as Author response images1 and 2. We would like to seek advice from the *eLife* editors and reviewers as to whether those images should be moved to the manuscript as additional supplementary figures.

Next, we will describe in detail the new experiments performed during the revision period to address validation and quantification of cortical fate:

1) We performed immunostaining on neural progenitor cells from humans and nonhuman primates (chimpanzees and bonobos) with 2 cortical progenitor markers suggested by the reviewers (FOXG1 and PAX6), followed by cell quantification. We detected similar percentages of PAX6- and FOXG1-positive cells in all species (updated new Figure 1—figure supplement 1).

2) We also assessed the bulk RNA-sequencing data from NPCs for developing brain regional signatures. Using *brainImageR* tool (Linker et al., 2018), we calculated the enrichment of each differentially expressed gene (described on Supplementary file 1) within tissues of the developing brain. Genes upregulated in human NPCs exhibited similar regional enrichment to genes upregulated in *Pan* (chimpanzee and bonobo) NPCs, indicating that the differences observed in bulk RNA sequencing were not influencing broad fate specification between NPCs from *Homo* versus *Pan* (new Figure 1—figure supplement figure 1).

3) To assess differences in cell purity, single-cell sequencing was performed using 10X platform on NPCs derived from a human, bonobo, or chimpanzee sample. We used progenitor cells from human and NHPs that were frozen at the same passage as the ones used for the transplantation experiments.

To determine whether there was a difference in the cortical layer identity between species, we referred to Molyneaux et al., 2007 for markers for each cortical layer: FOXO1, ECPN, LIX1, SYT9, S100A10, OMA1, LDB2, CRIM1, PCP4, RAC3, CITP2 and DIAP3 were used as markers for cortical layer Vb; LXN, FOXP2, DARPP32, and IGH6 for layer VI, *RORB*, CYP39A1, LHX2, UNC5D, GPR6, MEF2C, DTX4, CUX1, CUX2, KITL, and SVETL for layer II/III; and EN2, PAX2, EN1, ISL1, and LHX3 were included for hindbrain markers. Each species showed similar proportions of layer markers, indicating that within the cortex there was no particular preference for a given layer between the species (Author response image 1).

**Author response image 1. respfig1:** Proportion of cells expressing known markers for different cortical layers as well as hindbrain markers. Proportions are expressed out of the total number of NPCs expressing at least 1 marker.

2) Looking at the differentially expressed (DE) genes, numerous hits relate to hindbrain identity, suggesting significant biases during in vitro differentiation. As a result, the authors may be identifying forebrain versus non-forebrain cells. In line with this, Figure 1C-E show differentially expressed genes between human and pan. However, browsing the table of DE genes suggests a potentially problematic purity difference. 14 HOX genes are among the most significant DE genes and all show the same direction change (apparently up in pan according to the supplementary file). These genes are not expressed in the mammalian cortex, suggesting that the pan lines have a differentiation bias towards hindbrain or spinal cord fates. Without access to the full expression table, it is difficult to comment on how pervasive this issue is across all individuals versus a subset. However, this result raises a flag about the purity of the cells that needs to be addressed in order to understand whether the observed enriched gene categories relate to species differences or differences in cell purity.

To assess differences in cell purity, single-cell sequencing was performed on NPCs derived from a human, bonobo, or chimpanzee sample. Our first goal was to assess whether HOX expression was indicative of differing hindbrain-specifying NPCs between the species. To answer this question, we first identified all NPCs expressing SOX2, ensuring their NPC identity (Author response image 2; left column-dashed circle). Within the SOX2-expressing cells we next identified expression of one of the top differentially expressed HOX genes, HOXA7 (Author response image 2; middle column-dashed circle). Importantly, the majority of SOX2-expressing NPCs in both the bonobo and chimpanzee samples expressed HOXA7 whereas not a single human NPC expressed HOXA7. This finding indicated that HOXA7 expression was not marking heterogeneity within the NPC population but, as noted in bulk RNA-sequencing, was specific to the *Pan* species.

We next asked whether HOXA7 expression was indicative of NPCs being differentially primed toward the hindbrain or spinal cord fates across species. Using PAX6 and EN2 as markers of cortical and hindbrain NPCs, respectively, we noted that HOXA7 cells preferentially also expressed PAX6 (bonobo = 6.0%, chimpanzee = 8.2%) in comparison to EN2 (bonobo = 0%, chimpanzee = 0.7%). Importantly, the existence of NPCs that were double positive for HOXA7 and PAX6 in bonobos and chimpanzees and the striking absence of HOXA7 expression in any human NPCs further supported that HOXA7 expression was an intrinsic characteristic of *Pan* NPCs, independent of regional identity.

To determine whether this finding that human and *Pan* NPCs were similarly specified to cortical and hindbrain fates, independent of HOX expression, we determined the proportion of NPCs expressing well-known cortical and hindbrain markers (Author response image 2). SOX2 and NESTIN expression were used to identify NPCs; PAX6, NEUROD6 and TBR1 to identify cortical NPCs; and EN1, EN2, ISL1, and LHX3 to identify hindbrain NPCs. We identified similar proportions of all NPCs, cortical NPCs, and hindbrain NPCs in all species. This analysis further indicated that there was no detectable species-dependent bias in regional patterning.

Lastly, as mentioned in comment #1, we sought to determine whether there was a difference in the cortical layer identity between species (Author response image 1). We referred to Molyneaux et al., 2007 for markers for each layer. FOXO1, ECPN, LIX1, SYT9, S100A10, OMA1, LDB2, CRIM1, PCP4, RAC3, and DIAP3 were used as markers for cortical layer Vb; LXN, FOXP2, DARPP32, and IGH6 for layer VI, RORB, CYP39A1, LHX2, UNC5D, GPR6, MEF2C, DTX4, CUX1, CUX2, KITL, and SVETL for layer II/III; and EN2, PAX2, EN1, ISL1, and LHX3 were included for hindbrain markers. Again, each species showed similar proportions of layer markers, indicating that within the cortex there was also no particular preference for a given layer between the species (Author response image 1).

Together these results indicated that there was no substantial difference in broad fate specification between NPCs from human versus *Pan spp*.

**Author response image 2. respfig2:** Single cell analysis on human and *Pan* NPCs. (**A**) T-SNE of NPCs from human (top; green), bonobo (middle; orange), and chimpanzee (bottom; red). Populations that expressed SOX2, SOX2 and HOXA7, or all four markers, are noted in dashed circles. Percentage of PAX6+ or EN2+ cells out of all HOXA7-expressing NPCs are noted below the respective ovals. (**B**) Percentage of NPCs expressing known cortical or hindbrain markers for each species (human = green, bonobo = orange, chimpanzee = red). The existence of NPCs that were double positive for HOXA7 and PAX6 in bonobos and chimpanzees and the striking absence of HOXA7 expression in any human NPCs further supported that HOXA7 expression was an intrinsic characteristic of *Pan* NPCs, independent of regional identity.

3) The authors should make clear how many individuals were used for each experiment, and when statistical analyses are performed on technical replicates. This must be articulated explicitly, ideally with a table of individuals and lines that went into each experiment.

We articulated the information more explicitly in the text and provided a new table (Supplementary file 3) of individuals and lines that went into each experiment.

4) For the migration assay in vivo, it would be useful to have low magnification images of multiple sections to get a sense for the general distribution of cells within the section, relative to the site of injection and across animals.

We now provide low magnification images from an additional animal showing human and chimpanzee cells migrating. We highlighted the injection site (new Figure 3—figure supplement 1).

5) The migration assays in Figure 2 and the microelectrode array assays do not have a control for the purity of cells entering the assay. Cells from different regions may have different migratory properties and functional maturation rates that could influence the observed phenotypes. Post-staining representative sections showing that comparable populations were analyzed would help alleviate these concerns.

We further characterized the purity of the cells entering the migration and MEA assays by staining for neuroprogenitor cell markers (SOX2 and Nestin, Figure 1) and by staining for cortical progenitor markers (PAX6 and FOXG1, new Figure 1—figure supplement 1). There were no significant differences in the percentage of cells expressing cortical markers between human and NHP NPCs prior to migration assays or MEA.

To evaluate if differential migratory properties were influenced by the presence of cells from other brain regions due to loss of cell purity during migration, we post-stained human and NHP cells after a neurosphere migration experiment, as suggested by the reviewer. We observed that the percentages of cells expressing cortical progenitor markers (ex. PAX6) and neuroblast migration markers (ex. doublecortin, DCX) were not significantly different between species, even though the migration distances covered by human NPCs were significantly shorter than the distances covered by NHP cells (updated Figure 2—figure supplement 1K, L and M).

6) In their transplantation experiments, the authors don't provide enough insights into the types of cells that are compared. The only evidence shown is immunohistochemistry for CTIP2, but that marker only identifies a subset of cortical neurons, and may be more broadly expressed beyond cerebral cortex. Further characterization of these cells post transplantation would solve this problem.

We investigated the presence of other cortical markers on transplanted cells as suggested by the reviewers. We have performed immunostaining markers of other upper cortical layers such as Satb2 (Layers II and III) and Cux1 (cortical layers II-IV) and have not seen any co-localization from either human or NHP cells. We have now updated Figure 4—figure supplement 1 (D) to include images showing lack of co-localization with SATB2 cortical marker.

7) For the analysis of dendritic morphology, 3D reconstructions of dendritic arbors traced in their entirety in a volume of tissue would be very important. Adding images from multiple neurons would give the reader a better sense for the differences observed and their magnitude.

We agree with the reviewer’s point. As specified in the manuscript, we traced neurons in 3D using Neurolucidasoftware(MBF Bioscience, Williston, VT), as standard in neuroanatomical research. We added traced images from multiple neurons to the revised version of the manuscript (Figure 4—figure supplement 2J).